# Qualitative and Nutritional Evaluation of Paddlefish (*Polyodon spathula*) Meat Production

Daniel Simeanu [1], Răzvan-Mihail Radu-Rusu [2,*], Olimpia Smaranda Mintas [3,*] and Cristina Simeanu [2]

[1] Department of Control, Expertise and Services, Faculty of Food and Animal Sciences, "Ion Ionescu de la Brad" University of Life Sciences, 8 Mihail Sadoveanu Alley, 700489 Iasi, Romania

[2] Department of Animal Resources and Technology, Faculty of Food and Animal Sciences, "Ion Ionescu de la Brad" University of Life Sciences, 8 Mihail Sadoveanu Alley, 700489 Iasi, Romania

[3] Department of Livestock and Agrotourism, Faculty of Environmental Protection, University of Oradea, 1 University Street, 410087 Oradea, Romania

* Correspondence: radurazvan@uaiasi.ro (R.-M.R.-R.); olimpia.mintas@uoradea.ro (O.S.M.)

**Abstract:** *Polyodon spathula* is a valuable species of sturgeon native to North America that has acclimatized very well in Europe. Detailed knowledge of the quantitative and qualitative productive performance of paddlefish meat is of interest. Through this article, we aimed to highlight the chemical composition, cholesterol, and collagen content of fillets issued from paddlefish aged two and three summers and to highlight, as well as the nutritional value, the profile of fatty acids and amino acids, the sanogenic indices and the biological value of proteins for the epaxial and hypaxial muscle groups. The chemical analysis of the fillets by age indicated slightly higher values in summer three, compared to summer two: +5.32% dry matter, +0.89% protein, +41.21% fat, therefore +10.94% gross energy and for collagen by 2.94%; instead, for water, minerals and the W/P ratio the values were lower by 1.52%, 10.08%, and 2.29%. The nutritional assessment revealed that paddlefish has a meat with high PUFA content (approx. 22% of total fatty acids) and good values of sanogenic indices (Polyunsaturation Index = 7.01–8.77; Atherogenic Index = 0.57; Thrombogenic Index = 0.38–0.39; Hypocholesterolemic Fatty Acids = 33.01–41.34; Hypocholesteromic/Hypercolesteromic Fatty Acids ratio = 1.9). Also, the proteins of these fish are of good quality for young and adult consumers (EAA index = 156.11; Biological Value = 158.46; Nutritional Index (%) = 28.30) and good enough for children (Essential Amino Acids Index = 96.41; Biological Value = 93.39; Nutritional Index (%) = 17.45).

**Keywords:** paddlefish; meat quality; chemical composition; fatty acids; biological value

## 1. Introduction

Only one species of sturgeon from the *Polyodontidae* family is found in the waters of the North American continent, namely the *Polyodon spathula* or the commonly named paddlefish, spoon-billed cat, or spoonbill (Walbaum, 1972) [1]. These fish are distributed in the Mississippi River basin, from the Great Lakes area down to Florida. They can reach approx. 1.5–2 m body length and a body mass of about 50–70 kg upon maturity. Morphologically, *Polyodon spathula* is close to other sturgeons, with the exception of the head area where there is an elongated, dorsally-ventrally compressed rostrum that has the shape of a paddle. The length of this rostrum can reach 1/3 of the total body size in adult specimens. In this species, tactile and electrical receptors were found, deployed as a network on the operculum and the rostrum [2–6].

The sturgeon species *Polyodon spathula* usually does not have scales. However, small scales have been identified in the area of the lateral line, at the base of the dorsal fin and at the base of the pectoral fins [7,8]. The skin is smooth and its color varies from light gray to black on the dorsal area and on the flanks, and grayish-white on the abdomen. The paddle fish has poorly developed vision; the eyes are small and placed at the base of the rostrum [9,10]. The skeleton presents bone pieces only in the head area (jaws),

while the rest is cartilaginous. This fish has a well-developed filtering apparatus, being a planktonophagous species. The mouth is large, located in a ventral position, and the digestive tube has a formation specific to the *Acipenseriform* family, namely the spiral valve [11–13]. Paddlefish consume planktonic organisms—especially zooplankton, aquatic insects and larvae [14–17]. They reach sexual maturity at the age of 7–8 years in males and 10–13 years in females [18,19]. The eggs are small, with a diameter of 2.5–3 mm, and females lay about 100,000 eggs per reproduction season [20–22].

Reproduction takes place in spring, when the water temperature reaches 11–14 °C. Fish migrate even hundreds of kilometers in the upper part of the river or on its affluents [23–26]. Young sturgeons grow very fast in the first year of life, even during the winter period [11,27], after which the development becomes slower. The growth rate is around 5 cm body length/year throughout the first five years of life. Growth of body mass increases after this age, becoming double or even triples at 10 years old [28–31].

The freshwater sturgeon species, *Polyodon spathula*, has been important and acclimatized in Eastern Europe since 1992. At the moment, in Romania, thanks to the studies carried out by researchers from the Research and Development Station for Fisheries in Nucet and from the Research and Development for Aquaculture and Aquatic Ecology, Ciurea, Iași, it was facilitated the development of polyculture technologies, along with other species, such as the common carp and Asian cyprinids and also techniques for reproduction, pre-development and growth of young fish in protected areas, throughout the first year of life [32,33]. The success achieved in the research facilities also encouraged farmers to turn to this species of sturgeon for rearing for meat production. Thus, the qualitative meat production of these sturgeons, in general, but especially those obtained outside the usual breeding area—North America, must be studied in particular. *Polyodon spathula* meat has a taste and texture similar to that of other sturgeons, firm, white in color and boneless [34–37]. The carcasses of paddlefish have a "bullet" shape (no head, visceral mass and no fins) and represent 57% of the fish body. If the processing is carried out in the form of a fillet, the dressed yield decreases to 27% after removal of the skin and red meat [38,39]. Sturgeons' meat has been known and accepted since the beginning of the colonization of the North American continent in the late 1800s, but the meat of *Polyodon spathula* was less accepted. Currently, this meat is marketed under the title of "boneless catfish", to associate this meat with a popular product in the southern United States of America. As a result, the paddlefish meat still remains unknown to many consumers, and the market is very limited [40]. Meat from *Polyodon spathula* contains small amount of lipids, 1 to 4.5%, depending on the body mass and age of the individuals [38]. Concerning total protein content, the flank muscles (fillet) contain between 18 and 20% protein, placing this fish in the class of fish with a high protein content [41–43]. The meat has a stable shelf life since it can be kept refrigerated for up to seven days or even up to seven months under freezing conditions [38].

There are some issues when approaching the assessment of paddlefish meat production, both quantitatively and as quality, due to the fact that this species is reared mostly in polyculture in Romania and in small flocks, there are no studies on consumers' acceptability level and preferences. Also, the technological aspects at the fishery farm level are more difficult to manage, in comparison with other sturgeon species (water surface covering with nests to protect the paddlefish against its predators—waterfowl, difficulties to provide supplemental feeding to other species in polyculture), and farmers prefer to not choose paddlefish when populate the ponds, despite its natural good potential for good feed conversion weight gain using mostly natural feed sources (mostly zooplankton).

Within the presented context, our findings contribute to the most detailed knowledge of the *Polyodon spathula* sturgeon meat and to the completion of the scientific literature with original data on the most detailed on the chemical composition of the fillet, of the main muscles groups, and on the biological value of the proteins issued from paddlefish aged second and third summers raised in a fish farm in Romania.

## 2. Materials and Methods

### 2.1. Biological Material

A total of 500 sturgeons, *Polyodon spathula* species (300 individuals aged 2 Summers and 200 individuals aged 3 Summers) were used as main study populations. Fish were farmed to be sold on market in Hudesti Fishery, Botosani County, located in the North-East of Romania, at these geo coordinates: lat. N 48.1212, long. E 26.5519. The sturgeons were reared throughout the 2010–2012 timespan in a pool with a surface of 30 hectares, in polyculture with certain indigenous and Asian cyprinid species: *Cyprinus carpio* (indigenos carp, 39% of the fish in the pond), *Hypophtalmichthys molitrix* (Asian silver carp, 30% of the fish in the pond), *Ctenopharingodon idella* (Asian grass carp, 18% of the fish in the pond), *Aristichthys nobilis* (Asian bighead carp, 12% of the fish in the pond).

No supplemental feeding was designed and provided for paddlefish, knowing the species fed mostly through filtration and its natural feed source consists in zooplankton (and mostly in cladocerans). Concentrated feed was provided for other fish species in the pond and secondarily, it served to better develop the zooplankton eaten by paddlefish. The diets supplementarily fed to cyprinids consisted in a mixture of soybean meal, sunflower meal, corn, oat, and wheat flours and the proportions were optimized to achieve different nutritional levels (Table 1) in accordance with season and pond richness in natural resources.

**Table 1.** Nutritional features of the supplemental feed provided in fishery, throughout May-October, 2010–2012.

| Month | Metabolizable Energy kcal/kg | Crude Protein % | Lysine % | Methionine + Cystine % | Crude Fat % | Crude Fiber % | Ca % | P % |
|---|---|---|---|---|---|---|---|---|
| May | 2645 | 18.06 | 0.69 | 0.74 | 1.56 | 10.83 | 0.14 | 0.72 |
| June | 2617 | 20.53 | 0.76 | 0.83 | 1.33 | 11.88 | 0.16 | 0.82 |
| July | 2603 | 21.74 | 0.83 | 0.86 | 1.22 | 12.39 | 0.17 | 0.87 |
| August | 2600 | 22.00 | 0.84 | 0.89 | 1.20 | 12.50 | 0.18 | 0.88 |
| September | 2831 | 22.94 | 0.87 | 0.90 | 1.03 | 9.49 | 1.17 | 0.82 |
| October | 2893 | 26.34 | 1.15 | 0.99 | 1.16 | 9.92 | 0.19 | 0.84 |

In order to better depict the environmental conditions provided to policultural rearing of the paddlefish, Table 2 displays the water quality traits (average values) throughout the warm seasons of 2010–2012.

**Table 2.** Water quality traits in the paddlefish rearing pond.

| Water Trait | May | June | July | August | September |
|---|---|---|---|---|---|
| Colour | Green brown | Green | Green | Yellow green | Yellow green |
| Transparency (cm) | 24.58 | 19.67 | 18.34 | 16.21 | 17.76 |
| Temperature (°C) | 18.86 | 20.73 | 23.84 | 25.72 | 24.65 |
| Oxygen (mg/L) | 7.35 | 6.65 | 4.47 | 5.19 | 5.88 |
| pH value (pHU) | 7.39 | 7.23 | 7.17 | 6.51 | 6.75 |
| Calcium (mg/L) | 52.54 | 60.78 | 56.76 | 46.94 | 40.75 |
| Magnesium (mg/L) | 25.42 | 28.31 | 19.78 | 18.10 | 18.34 |
| Chlorides (mg/L) | 17.98 | 12.45 | 11.78 | 13.12 | 15.32 |
| Nitrates (mg/L) | 0.32 | 0.24 | 0.14 | 0.16 | 0.16 |
| Nitrites (mg/L) | 0.32 | 0.27 | 0.26 | 0.18 | 0.16 |
| Phosphates (mg/L) | 0.02 | 0.02 | 0.03 | 0.03 | 0.03 |
| Organic matters (mg/L) | 68.7 | 71.6 | 79.8 | 98.5 | 59.6 |

Out of the 500 sturgeons in the pond, upon harvesting for selling them on the market, 10% were weighted and 20 fishes per each age (summer two and summer three) were chosen for meat sampling after refrigeration and analytical laboratory investigations. No animals were used for applying experimental factors and to measure their effect throughout certain specific reasoning criteria. The specimens used for meat sampling were raised in

common with the other marketable fishes and the samples were taken after the exitus was installed in refrigeration marketing condition.

From the specimens chosen for analysis, samples were collected from in the form of fillets (50% of fish) and from the groups of lateral muscles (50% of fish). Each lateral muscle is divided along its length into two masses, one dorsal (epaxial) and one ventral (hypaxial), by a connective wall (transverse myoseptum) fixed to the integument and the axial skeleton and located slightly below the lateral line. The muscles located above the lateral line are called epaxial muscles and are divided into dorsal epaxial muscles and costal epaxial muscles, while those located below the lateral line are called hypaxial muscles and are divided into costal hypaxial muscles and abdominal hypaxial muscles [1] (Figure 1).

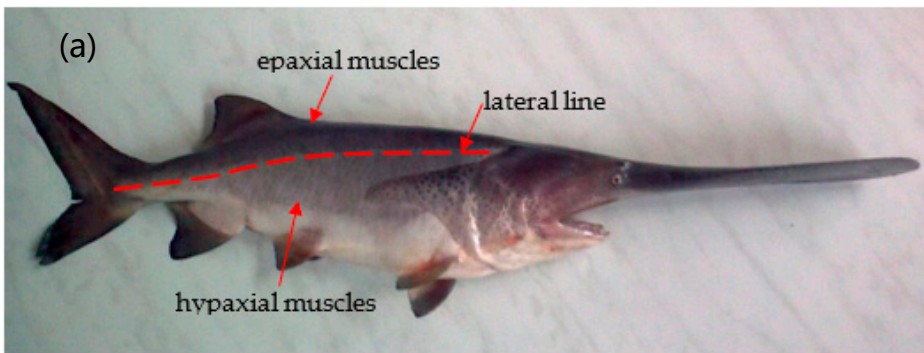

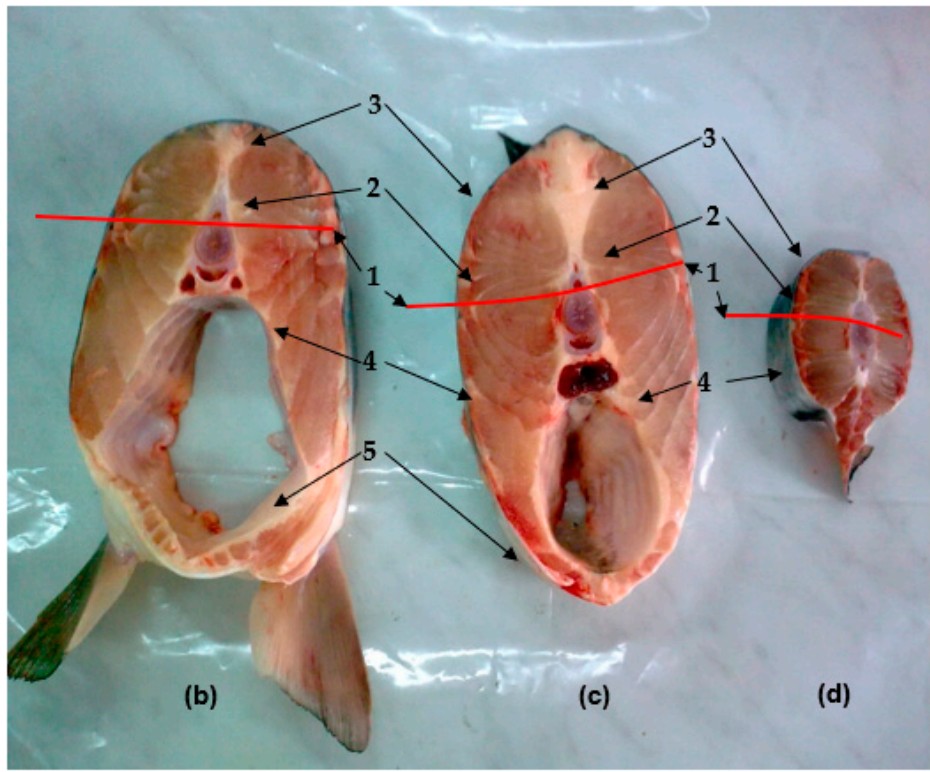

**Figure 1.** Side muscles in *Polyodon spathula*: (**a**) topographic placement; (**b**,**c**)—cross sections through the abdominal area; (**d**)—cross section through the caudally area; 1—side line; 2—costal epaxial muscles; 3—dorsal epaxial muscles; 4—costal hypaxial muscles; 5—abdominal hypaxial muscles.

*2.2. Assessment of Proximate Chemical Composition and Energetic Value*

Water was assessed by oven dehydration method (drying of sample at +105 °C temperature) following the SR ISO 1442/1997 analytical protocol.

Total minerals were measured gravimetrically by calcinations method in electric furnace, using a working temperature of +550 °C (SR ISO 936:1998 standard).

Protein was assessed via Kjeldhal method, carried on in accordance with the 981:10; AOAC Official methods of analysis/1990, compatible with SR ISO 937:2007, adapted to Velp Scientifica analytical devices (DK6 digester; UDK7 distiller) (manufacturer VELP Scientifica SRL, Usmate Velate, Italy) [43–46].

Water/protein ratio (W/P) was calculated by reporting the water analytical content of each sample to the crude protein content of each sample.

Total lipids were measured on the Velp Scientifica–SER 148 extractor (manufacturer VELP Scientifica SRL, Usmate Velate, Italy), using the Soxhlet method (AOAC Officinal methods of analysis/1990 [43–46]), compatible with the SR ISO 1443:2008 protocols [43,47].

Nitrogen free extract (remains of carbohydrates not analyzed individually) was calculated algebraically, by difference between the total solids (dry matter) content, total minerals content, total proteins content, and total lipids content.

Meat gross energy (GE) content was derived through mathematical computation, using the raw calories content of one grams of each organic matter component when burnt into a calorimeter, in accordance with the relation (1) [42,43]:

$$GE \text{ (kcal/100 g)} = g\ TP \times 5.70\ \text{kcal} + g\ TL \times 9.50\ \text{kcal} + g\ NFE \times 4.2\ \text{kcal} \qquad (1)$$

where: GE = gross energy, TP = total proteins, TL = total lipids, NFE = nitrogen free extract.

Omega Bruins Food-Check Near InfraRed (NIR) spectrophotometer (manufacturer Bruins Instruments GmbH, Puchheim, Germany) was used to run rapid scanning transmission tests for collagen content assessment [42,43].

Each sample was analyzed in 30 repetitions, for proximate composition, collagen content and 30 repetitions have been carried out for W/P ratio and GE calculations.

## 2.3. Fatty Acids Profile Analysis and Lipids Nutritional Quality

Fatty acid methyl esters (FAME) in paddlefish meat were extracted and quantified by gas chromatography and mass spectrometry detection, using the Perkin Elmer Chromatographic system coupled with mass spectrometer detector (GC-MS) (Clarus 680 gas-chromatograph and Clarus SQ8T quadrupole mass spectrometer (manufacturer, Perkin Elmer Inc., Boston, MA, USA, for both devices). An elite-wax with stationary polar phase Polyethylene glycol (PEG) chromatographic column was used (total length 30 m, 1.0 μm film thickness and internal diameter of 0.25 mm, injecting port temp. = 220 °C, injected sample = 1.0 μL, He carrier flew at 1.5 mL/min rate, while used splitting ratio was 40:1). Temperature gradient was set at 100 °C, throughout 2 min, standstill, and stationary 1 min at 250 °C. Mass Spectrometer was characterized by the following operational values: 150 °C the temperature of the transfer line; 150 °C the temperature of the source; 1500 multiplier; solvent delay of 0–1.5 min for the solvent [48–51].

The method uses the chromatographic isolation of the fatty acid melange in samples, after esterifying them with $CHNaO_2$ (sodium formate or methanoate), on a capillary column, and fragmenting the molecules by ionization. Identification and measuring of FAME quantities are carried on using target and qualifying ions. The FAME in *Polyodon spathula* meat issued from prior lipid saponification phase, followed by boron trifluoride (15% vol.) catalyze on the esterification reaction. Fatty acids values in the samples were achieved by running the comparison between FAME retention time with a homologated standard (FAME Supelco 37 Mix). Each fatty acid was quantified as g FAME/100 g of total identified FAME [48–51].

Seven analytical repetitions were carried out per sample to quantify the fatty acids and cholesterol content in paddlefish meat.

To outline the lipid profile, the fatty acids were grouped accordingly: saturated fatty acids–SFA, as sum (Σ SFA = C10:0 + C12:0 + C14:0 + C15:0 + C16:0 + C18:0 + C20:0 + C22:0); mono unsaturated fatty acids–MUFA, as sum (Σ MUFA = C16:1 + C18:1 *cis*−9 + C20:1 n−9 + C22:1 n−9); poly unsaturated fatty acids, as sum ((Σ PUFA = C18:2 n−6 + C20:2 n−6 + C18:3 n−3 + C20:3 n−3 + C20:5 n−3 + C18:3 n−6), total unsaturated fatty acids, as sum of MUFA and PUFA [49,50,52].

Quantities of Omega-3 and Omega-6 PUFA series were expressed as ration (n−3/n−6). Paddlefish meat polyunsaturation index (PI) was calculated by using the relation (2), published by Timmons [49,52,53]:

$$PI = C18:2 \, n-6 + (C18:3 \, n-3 \times 2). \tag{2}$$

The atherogenic index (AI) and thrombogenic index (TI) of fats were calculated, using the data issued from FAME GC analysis of the paddlefish meat, applying the relations (3) and (4) published by Ulbricht and Southgate [48,52,54,55]:

$$AI = (C12:0 + C16:0 + 4 \times C14:0)/[\Sigma \, MUFA + \Sigma \, (n-6) + \Sigma \, (n-3)], \tag{3}$$

$$TI = (C14:0 + C16:0 + C18:0)/[0.5 \times \Sigma \, MUFA + 0.5 \times \Sigma \, (n-6) + 3 \times \Sigma \, (n-3) + \Sigma(n-3)/\Sigma(n-6)]. \tag{4}$$

The relation (5) proposed by Fernandez et al. [55] was used to calculate the ratio between fatty acids with hypocholesterolemic (h) and hypercholesterolemic (H) effect.

$$h/H \, (hypocholesterolemic/Hypercholesterolemic) = (C18:1 + PUFA)/(C12:0 + C14:0 + C16:0). \tag{5}$$

### 2.4. Amino Acid Analysis and Protein Quality Assessment

Amino acids in paddlefish meat were identified and quantifies using the Model 118/119 CL, CR Mikrotechna AAA 881 automatic analyzer (manufacturer Mikrotechna, Prague, Czech Republic). Samples were hydrolyzed in HCl 6 M at 110 °C, continuously throughout 24 h, using nitrogen atmosphere. Cysteine and methionine were assessed in form of cysteic acid and methionine sulfone, following sample oxidation with performic acid. Tryptophan was assessed after NaOH hydrolysis at 110 °C, throughout 22 h, in accordance with the protocol specified by Official Methods of Analysis of the Association of Analytical Chemists [44,46]. Amino acid quantification was expressed in the g/16 g N system, equivalated to g/100 g of total protein [43,56–58].

Seven analytical repetitions were carried out per sample to quantify the Amino Acids.

Quality of the proteins was assessed by methods using parameters such as: inner content in essential amino acids (AA), fraction of amino acids of exogenous origin (EAA), protein chemical score (CS) (relation 6) and the essential amino acid index (EAAI) (relation 7) [43,48,58,59].

$$CS = \frac{\text{content in amino acid A in the studied protein}}{\text{content in amino acid A the standard protein}} \times 100 \tag{6}$$

Usually, egg protein is used as standard or etalon protein, in accordance with certain reference international scientific bodies: Food and Agriculture Organization of the United Nation (FAO)/World Health Organization (WHO)/United Nations University (UNU). Nutritional value of proteins was computed, in this research, according to the standard requirements:

- Standard 1—children consumers: Tryptophan = 1.7; Threonine = 4.3; Isoleucine = 4.6; Leucine = 9.3; Lysine = 6.6; Methionine + Cysteine = 4.2; Phenylalanine + Tyrosine = 7.2; Valine = 5.5; EAA = 37.9 g/16 g Nitrogen) [56–58,60]
- Standard 2—youth consumers: Tryptophan = 1.0; Threonine = 4.0; Isoleucine = 4.0; Leucine = 7.0; Lysine = 5.5; Methionine + Cysteine = 3.5; Phenylalanine + Tyrosine = 6; Valine = 5; EAA = 36 g/16 g Nitrogen) [56–58,61]
- Standard 3—adult consumers: Tryptophan = 0.6; Threonine = 2.6; Isoleucine = 3; Leucine = 4.4; Lysine = 3.1; Methionine + Cysteine = 2.7; Phenylalanine + Tyrosine = 3.3; Valine = 2.3; EAA = 22 g/16 g Nitrogen) [56–58,61]

The EAAI (essential amino acid index) was calculated using relation (7) and the values the chemical indexes for the eight essential amino acids, in accordance to Oser's methodology [43,48,58,62]:

$$EAAI = \sqrt[n]{CS1 \times CS2 \times CS3 \times \ ... \ \times \ CSn} \tag{7}$$

Oser's [48,58,63] mathematical relation (8) was applied to compute the proteins Biological Value (BV):

$$BV = 1.09 \ (EAAI) - 11.7. \tag{8}$$

The mathematical relation (9) proposed by Crisan and Sands, 1978 [43,48,58,64] was used to compute the nutritional index (NI) of paddlefish meat:

$$NI \ (\%) = \frac{EAAI \ \times \ \% \ protein}{100}. \tag{9}$$

### *2.5. Data Analysis*

The analytical data were input in a Microsoft Excel 2016 database then processed to calculate main statistic descriptors (mean, variance, standard deviation, standard error of mean, coefficient of variation), and analysis of variance—one-way ANOVA, followed by post hoc Tukey testing—using the GraphPad Prism 9.4.0 (673) software (manufacturer, GraphPad Software, San Diego, CA, USA).

Statistics outputs are based on 30 analytical repetitions in proximate content, W/P ratio, collagen and gross energy and on 7 analytical repetitions for fatty acids, cholesterol and amino acids contents.

## 3. Results

### *3.1. Chemical Proximate Composition*

Results of the proximate chemical analysis of *Polyodon spathula* sturgeon fillets from the 2nd and 3rd summer are displayed in Table 3, along with data on the gross energy value, collagen content and water/protein ratio.

**Table 3.** Chemical composition and certain nutritional features of *Polyodon spathula* fillet, aged two and three summers.

| Age | Statistics | Water | Dry Matter | Protein | Total Fat | Minerals | Energy | Collagen | W/P Ratio |
|---|---|---|---|---|---|---|---|---|---|
| S2 | Mean ± SD | 77.83 ± 0.27 | 22.17 ± 0.27 | 17.94 ± 0.18 | 2.82 ± 0.14 | 1.41 ± 0.30 | 102.05 ± 0.81 | 3.93 ± 0.06 | 4.34 ± 0.05 |
|  | CV | 0.35 | 1.21 | 0.99 | 5.02 | 21.13 | 0.79 | 1.61 | 0.09 |
| S3 | Mean ± SD | 76.65 ± 0.25 | 23.35 ± 0.25 | 18.10 ± 0.13 | 3.98 ± 0.12 | 1.27 ± 0.34 | 113.22 ± 1.32 | 4.04 ± 0.08 | 4.24 ± 0.03 |
|  | CV | 0.33 | 1.07 | 0.74 | 3.12 | 26.82 | 1.17 | 1.86 | 0.79 |
| Comparisons | ±% S3 vs. S2 | −1.52% | +5.32% | +0.89% | +41.21% | −10.08% | +10.94% | +2.94% | −2.29% |
|  | *p* values | $2.53 \times 10^{-10}$ | $2.53 \times 10^{-10}$ | 0.9928 | $2.53 \times 10^{-10}$ | 0.9979 | $2.53 \times 10^{-10}$ | 0.9998 | 0.9998 |

S2—2nd summer. S3—3rd summer SD—standard deviation. CV—coefficient of variation. *n* = 30.

Same parameters were analysed separately per muscular groups (epaxial dorsal, epaxial costal, hypaxial costal and hypaxial abdominal) on samples taken from paddlefish aged two summers (Table 4).

Chemical analysis results from the same muscle groups issued from *Polyodon spathula* sturgeons aged three summers are presented in Table 5.

Achieved data was compared between the two investigated ages, both as relative differences (± %) and as analysis of variance (Table 6).

### *3.2. Fatty Acids Profiling and the Sanogenic Indices of the Polyodon Spathula Sturgeons Meat*

Profile of fatty acids and the sanogenic indices assessment are presented in Table 7. The analysis was carried on for each age and on epaxial and hypaxial muscles groups.

**Table 4.** Chemical composition and certain nutritional features of *Polyodon spathula* meat, aged two summers.

| Muscles | Descriptive Statistics | Water (g/100 g) | Dry Matter (g/100 g) | Protein (g/100 g) | Total Fat (g/100 g) | Minerals (g/100 g) | Energy (Kcal/100 g) | Collagen (g/100 g) | W/P Ratio |
|---|---|---|---|---|---|---|---|---|---|
| ED | Mean ± SD | 77.17 ± 0.46 | 22.83 ± 0.46 | 18.17 ± 0.30 | 3.29 ± 0.22 | 1.37 ± 0.41 | 107.28 ± 1.56 | 3.92 ± 0.12 | 4.25 ± 0.09 |
|  | CV | 0.59 | 2.00 | 1.65 | 6.55 | 30.20 | 1.46 | 3.99 | 2.07 |
| EC | Mean ± SD | 78.36 ± 0.62 | 21.64 ± 0.62 | 18.06 ± 0.29 | 2.19 ± 0.16 | 1.39 ± 0.62 | 96.88 ± 1.44 | 3.79 ± 0.12 | 4.34 ± 0.08 |
|  | CV | 0.80 | 2.16 | 1.63 | 7.10 | 45.02 | 1.48 | 3.18 | 1.86 |
| HC | Mean ± SD | 77.96 ± 0.37 | 22.04 ± 0.37 | 17.59 ± 0.27 | 2.98 ± 0.20 | 1.47 ± 0.35 | 101.99 ± 2.04 | 4.08 ± 0.05 | 4.43 ± 0.08 |
|  | CV | 0.48 | 1.70 | 1.56 | 6.62 | 33.46 | 2.00 | 1.33 | 1.79 |
| HA | Mean ± SD | 44.05 ± 0.42 | 55.95 ± 0.42 | 16.77 ± 0.31 | 38.15 ± 0.42 | 1.03 ± 0.68 | 415.71 ± 4.26 | 4.26 ± 0.08 | 2.63 ± 0.05 |
|  | CV | 0.95 | 0.75 | 1.84 | 1.11 | 65.58 | 1.02 | 1.81 | 1.94 |
| *p* values (ANOVA) | ED vs. EC | $4.10 \times 10^{-14}$ | $4.10 \times 10^{-14}$ | $3.63 \times 10^{-11}$ | $2.08 \times 10^{-15}$ | 0.9989 | $2.41 \times 10^{-15}$ | $3.74 \times 10^{-6}$ | $6.80 \times 10^{-5}$ |
|  | ED vs. HC | $2.21 \times 10^{-8}$ | $2.21 \times 10^{-8}$ | $2.15 \times 10^{-15}$ | 0.0001 | 0.8843 | $9.08 \times 10^{-12}$ | $9.36 \times 10^{-9}$ | $4.45 \times 10^{-14}$ |
|  | ED vs. HA | $2.15 \times 10^{-15}$ | $2.15 \times 10^{-15}$ | $4.92 \times 10^{-8}$ | $2.16 \times 10^{-15}$ | 0.0968 | $2.29 \times 10^{-15}$ | $2.27 \times 10^{-15}$ | $2.43 \times 10^{-15}$ |
|  | EC vs. HC | 0.0079 | 0.0079 | $2.15 \times 10^{-15}$ | $1.74 \times 10^{-14}$ | 0.9360 | $3.90 \times 10^{-11}$ | $9.05 \times 10^{-15}$ | $2.37 \times 10^{-5}$ |
|  | EC vs. HA | $2.31 \times 10^{-15}$ | $2.31 \times 10^{-15}$ | $2.37 \times 10^{-15}$ | $2.51 \times 10^{-15}$ | 0.0690 | $2.38 \times 10^{-15}$ | $2.46 \times 10^{-15}$ | $2.05 \times 10^{-15}$ |
|  | EC vs. HA | $2.19 \times 10^{-15}$ | $2.19 \times 10^{-15}$ | 0.9935 | $2.29 \times 10^{-15}$ | 0.0142 | $2.19 \times 10^{-15}$ | $6.96 \times 10^{-11}$ | $2.61 \times 10^{-15}$ |

ED—Epaxial dorsal. EC—Epaxial costal. HC—hypaxial costal. HA—Hypaxial abdominal. SD—standard deviation. CV—coefficient of variation. W/P—Water/Protein ratio. *n* = 30.

**Table 5.** Chemical composition and certain nutritional features of *Polyodon spathula* meat, aged three summers.

| Muscles | Descriptive Statistics | Water (g/100 g) | Dry Matter (g/100 g) | Protein (g/100 g) | Total Fat (g/100 g) | Minerals (g/100 g) | Energy (Kcal/100 g) | Collagen (g/100 g) | W/P Ratio |
|---|---|---|---|---|---|---|---|---|---|
| ED | Mean ± SD | 76.62 ± 0.50 | 23.38 ± 0.50 | 18.80 ± 0.34 | 3.37 ± 0.22 | 1.22 ± 0.59 | 110.64 ± 2.37 | 4.03 ± 0.09 | 4.08 ± 0.07 |
|  | CV | 0.66 | 2.16 | 1.79 | 6.48 | 48.69 | 2.14 | 2.29 | 1.81 |
| EC | Mean ± SD | 76.47 ± 0.48 | 23.53 ± 0.48 | 18.19 ± 0.28 | 4.09 ± 0.20 | 1.25 ± 0.59 | 114.56 ± 2.00 | 3.96 ± 0.14 | 4.21 ± 0.08 |
|  | CV | 0.63 | 2.05 | 1.54 | 4.78 | 46.95 | 1.74 | 3.61 | 1.82 |
| HC | Mean ± SD | 76.85 ± 0.43 | 23.15 ± 0.43 | 17.32 ± 0.24 | 4.49 ± 0.22 | 1.34 ± 0.59 | 114.44 ± 2.29 | 4.14 ± 0.07 | 4.44 ± 0.06 |
|  | CV | 0.86 | 1.86 | 1.36 | 5.01 | 44.26 | 2.00 | 1.58 | 1.42 |
| HA | Mean ± SD | 43.92 ± 0.37 | 56.09 ± 0.37 | 16.77 ± 0.31 | 38.22 ± 0.69 | 1.61 ± 0.70 | 414.16 ± 6.31 | 4.32 ± 0.05 | 2.70 ± 0.04 |
|  | CV | 0.85 | 0.66 | 1.84 | 1.82 | 43.55 | 1.52 | 1.13 | 1.48 |
| *p* values (ANOVA) | ED vs. EC | $2.09 \times 10^{-15}$ | $2.09 \times 10^{-15}$ | $1.97 \times 10^{-13}$ | $5.98 \times 10^{-10}$ | 0.9968 | 0.0004 | 0.0172 | $3.92 \times 10^{-11}$ |
|  | ED vs. HC | $2.84 \times 10^{-14}$ | $2.84 \times 10^{-14}$ | $2.61 \times 10^{-15}$ | $2.36 \times 10^{-14}$ | 0.8653 | 0.0006 | $3.85 \times 10^{-5}$ | $2.33 \times 10^{-15}$ |
|  | ED vs. HA | $2.28 \times 10^{-15}$ | $2.28 \times 10^{-15}$ | $2.42 \times 10^{-15}$ | $2.03 \times 10^{-15}$ | 0.0721 | $2.08 \times 10^{-15}$ | $7.12 \times 10^{-15}$ | $2.11 \times 10^{-15}$ |
|  | EC vs. HC | 0.0083 | 0.0083 | $4.16 \times 10^{-15}$ | 0.0007 | 0.9410 | 0.9992 | $2.67 \times 10^{-11}$ | $2.41 \times 10^{-15}$ |
|  | EC vs. HA | $2.11 \times 10^{-15}$ | $2.11 \times 10^{-15}$ | $2.63 \times 10^{-15}$ | $2.24 \times 10^{-15}$ | 0.1156 | $2.17 \times 10^{-15}$ | $2.17 \times 10^{-15}$ | $2.58 \times 10^{-15}$ |
|  | EC vs. HA | $2.25 \times 10^{-15}$ | $2.25 \times 10^{-15}$ | $2.17 \times 10^{-15}$ | $5.98 \times 10^{-10}$ | 0.3375 | $2.34 \times 10^{-15}$ | $2.76 \times 10^{-10}$ | $2.36 \times 10^{-15}$ |

ED—Epaxial dorsal. EC—Epaxial costal. HC—hypaxial costal. HA—Hypaxial abdominal. SD—standard deviation. CV—coefficient of variation. W/P—Water/Protein ratio. *n* = 30.

**Table 6.** Analysis of variance between ages (two summers vs. three summers) on the chemical composition and certain nutritional features of *Polyodon spathula* meat.

| Muscles | Percent Differences | Water | Dry Matter | Protein | Total Fat | Minerals | Energy | Collagen | W/P Ratio |
|---|---|---|---|---|---|---|---|---|---|
| ED | ±% S3 vs. S2 | −0.71% | +2.40% | +3.43% | −2.34% | −11.12% | +3.13% | +2.86% | −4.01% |
|  | *p* values | 0.2701 | 0.2701 | 0.1017 | 0.9999 | 0.9999 | $2.53 \times 10^{-10}$ | 0.9999 | 0.9999 |
| EC | ±% S3 vs. S2 | −2.41% | +8.72% | +0.70% | +86.76% | −10.09% | +18.25% | +4.49% | −3.09% |
|  | *p* values | 1.0167 | 1.0167 | 0.9999 | $2.53 \times 10^{-10}$ | 0.9999 | $2.53 \times 10^{-10}$ | 0.9999 | 0.9999 |
| HC | ±% S3 vs. S2 | −1.41% | +5.00% | −1.55% | +50.67% | −9.10% | +12.21% | +1.57% | +0.13% |
|  | *p* values | 1.0329 | 1.0329 | 0.9968 | $2.86 \times 10^{-10}$ | 0.9999 | $2.53 \times 10^{-10}$ | 0.9998 | 1.0064 |
| HA | ±% S3 vs. S2 | −0.32% | +0.25% | −3.04% | +0.18% | +56.27% | −0.37% | +1.57% | +2.80% |
|  | *p* values | 1.0428 | 1.0428 | 0.9998 | 1.2613 | 0.9991 | 0.1007 | 0.9999 | 0.9999 |

ED—Epaxial dorsal. EC—Epaxial costal. HC—hypaxial costal. HA—Hypaxial abdominal. S2—2nd summer. S3—3rd summer.

**Table 7.** Profile of fatty acids (g/100 g total FAME), cholesterol content (mg/100 g) and sanogenic indices of *Polyodon spathula* sturgeons meat, aged two and three summers.

| Fatty Acid | | Descriptive Statistics | Summer 2 | | Summer 3 | | p Values |
|---|---|---|---|---|---|---|---|
| | | | Epaxial Muscles | Hypaxial Muscles | Epaxial Muscles | Hypaxial Muscles | |
| Myristic | C14:0 | Mean ± SD<br>CV | 2.02 ± 0.06<br>2.81 | 2.16 ± 0.07<br>3.44 | 2.44 ± 0.07<br>2.76 | 2.53 ± 0.06<br>2.55 | ES3 vs. ES2, $p = 1.81 \times 10^{-14}$<br>HS3 vs. HS2, $p = 1.62 \times 10^{-13}$ |
| Pentadecanoic | C15:0 | Mean ± SD<br>CV | 0.40 ± 0.01<br>3.14 | 0.43 ± 0.01<br>2.86 | 0.49 ± 0.01<br>2.94 | 0.50 ± 0.02<br>4.27 | ES3 vs. ES2, $p = 1.57 \times 10^{-9}$<br>HS3 vs. HS2, $p = 1.74 \times 10^{-7}$ |
| Palmitic | C16:0 | Mean ± SDCV | 15.31 ± 0.21<br>1.35 | 16.38 ± 0.57<br>3.51 | 18.53 ± 0.59<br>3.18 | 19.18 ± 0.61<br>3.19 | ES3 vs. ES2, $p = 1.83 \times 10^{-14}$<br>HS3 vs. HS2, $p = 1.79 \times 10^{-14}$ |
| Palmitoleic | C16:1 | Mean ± SD<br>CV | 8.10 ± 0.26<br>3.27 | 8.67 ± 0.25<br>2.91 | 9.80 ± 0.36<br>3.69 | 10.14 ± 0.31<br>3.10 | ES3 vs. ES2, $p = 1.80 \times 10^{-14}$<br>HS3 vs. HS2, $p = 1.82 \times 10^{-14}$ |
| Margaric | C17:0 | Mean ± SD<br>CV | 0.67 ± 0.03<br>4.36 | 0.71 ± 0.03<br>4.08 | 0.80 ± 0.03<br>3.14 | 0.83 ± 0.02<br>2.42 | ES3 vs. ES2, $p = 7.56 \times 10^{-13}$<br>HS3 vs. HS2, $p = 4.17 \times 10^{-12}$ |
| Stearic | C18:0 | Mean ± SD<br>CV | 1.45 ± 0.04<br>2.84 | 1.55 ± 0.05<br>3.52 | 1.76 ± 0.05<br>2.72 | 1.82 ± 0.06<br>3.52 | ES3 vs. ES2, $p = 1.76 \times 10^{-14}$<br>HS3 vs. HS2, $p = 1.80 \times 10^{-14}$ |
| Oleic | C18:1 *cis*–9 | Mean ± SD<br>CV | 15.18 ± 0.55<br>3.61 | 16.24 ± 0.36<br>2.19 | 18.36 ± 0.82<br>4.44 | 19.01 ± 0.80<br>4.21 | ES3 vs. ES2, $p = 1.81 \times 10^{-14}$<br>HS3 vs. HS2, $p = 1.79 \times 10^{-14}$ |
| Asclepic | C18:1 *cis*–11 | Mean ± SD<br>CV | 4.30 ± 0.12<br>2.72 | 4.60 ± 0.20<br>4.27 | 5.21 ± 0.15<br>2.90 | 5.39 ± 0.16<br>2.91 | ES3 vs. ES2, $p = 1.80 \times 10^{-14}$<br>HS3 vs. HS2, $p = 1.82 \times 10^{-14}$ |
| Linoleic | C18:2, n–6 | Mean ± SD<br>CV | 1.71 ± 0.06<br>3.39 | 1.83 ± 0.05<br>2.62 | 2.07 ± 0.06<br>2.74 | 2.14 ± 0.06<br>2.82 | ES3 vs. ES2, $p = 1.78 \times 10^{-14}$<br>HS3 vs. HS2, $p = 1.82 \times 10^{-14}$ |
| α-linolenic | C18:3, n–3 | Mean ± SD<br>CV | 2.65 ± 0.11<br>4.08 | 2.83 ± 0.09<br>3.13 | 3.20 ± 0.12<br>3.80 | 3.32 ± 0.11<br>3.44 | ES3 vs. ES2, $p = 1.81 \times 10^{-14}$<br>HS3 vs. HS2, $p = 1.84 \times 10^{-14}$ |
| γ-linolenic | C18:3, n–6 | Mean ± SD<br>CV | 0.42 ± 0.01<br>3.54 | 0.45 ± 0.02<br>4.01 | 0.51 ± 0.02<br>3.24 | 0.53 ± 0.02<br>3.19 | ES3 vs. ES2, $p = 1.57 \times 10^{-9}$<br>HS3 vs. HS2, $p = 1.52 \times 10^{-8}$ |
| Arahic | C20:0 | Mean ± SD<br>CV | 0.09 ± 0.01<br>5.99 | 0.09 ± 0.01<br>5.82 | 0.10 ± 0.01<br>5.28 | 0.11 ± 0.01<br>4.92 | ES3 vs. ES2, $p = 0.6656$<br>HS3 vs. HS2, $p = 0.1282$ |
| Gadoleic | C20:1 | Mean ± SD<br>CV | 0.47 ± 0.01<br>2.76 | 0.50 ± 0.01<br>2.66 | 0.57 ± 0.03<br>4.80 | 0.59 ± 0.02<br>4.16 | ES3 vs. ES2, $p = 1.90 \times 10^{-10}$<br>HS3 vs. HS2, $p = 1.57 \times 10^{-8}$ |
| Eicosadienoic | C20:2 | Mean ± SD<br>CV | 0.29 ± 0.01<br>3.24 | 0.31 ± 0.01<br>2.37 | 0.35 ± 0.01<br>2.85 | 0.36 ± 0.01<br>3.22 | ES3 vs. ES2, $p = 2.45 \times 10^{-6}$<br>HS3 vs. HS2, $p = 3.74 \times 10^{-5}$ |
| Dihomo-γ-linolenic | C20:3, n–6 | Mean ± SD<br>CV | 0.12 ± 0.01<br>4.61 | 0.13 ± 0.01<br>4.15 | 0.15 ± 0.01<br>4.22 | 0.15 ± 0.01<br>4.61 | ES3 vs. ES2, $p = 0.0108$<br>HS3 vs. HS2, $p = 0.1282$ |
| Eicosatrienoic | C20:3, n–3 | Mean ± SD<br>CV | 0.34 ± 0.01<br>3.18 | 0.36 ± 0.01<br>3.32 | 0.41 ± 0.01<br>2.70 | 0.42 ± 0.01<br>2.72 | ES3 vs. ES2, $p = 1.74 \times 10^{-7}$<br>HS3 vs. HS2, $p = 2.37 \times 10^{-6}$ |
| Arachidonic | C20:4, n–6 | Mean ± SD<br>CV | 0.94 ± 0.02<br>2.35 | 1.01 ± 0.03<br>2.88 | 1.14 ± 0.04<br>3.47 | 1.18 ± 0.04<br>3.57 | ES3 vs. ES2, $p = 1.81 \times 10^{-14}$<br>HS3 vs. HS2, $p = 1.95 \times 10^{-14}$ |
| Eicosapentaenoic | C20:5, n–3 | Mean ± SD<br>CV | 3.01 ± 0.11<br>3.58 | 3.22 ± 0.10<br>3.16 | 3.64 ± 0.13<br>3.54 | 3.77 ± 0.12<br>3.11 | ES3 vs. ES2, $p = 1.80 \times 10^{-14}$<br>HS3 vs. HS2, $p = 1.77 \times 10^{-14}$ |
| Clupanodonic | C22:5, n–3 | Mean ± SD<br>CV | 1.75 ± 0.06<br>3.66 | 1.87 ± 0.08<br>4.05 | 2.11 ± 0.07<br>3.12 | 2.19 ± 0.07<br>3.18 | ES3 vs. ES2, $p = 1.81 \times 10^{-14}$<br>HS3 vs. HS2, $p = 1.79 \times 10^{-14}$ |
| Docosahexaenoic | C22:6, n–3 | Mean ± SD<br>CV | 2.31 ± 0.05<br>2.18 | 2.47 ± 0.09<br>3.67 | 2.80 ± 0.11<br>3.85 | 2.90 ± 0.11<br>3.76 | ES3 vs. ES2, $p = 1.82 \times 10^{-14}$<br>HS3 vs. HS2, $p = 1.77 \times 10^{-14}$ |
| Cholesterol | | Mean ± SD<br>CV | 48.86 ± 3.07<br>6.28 | 52.27 ± 1.42<br>2.71 | 59.10 ± 1.85<br>3.13 | 61.18 ± 2.18<br>3.56 | ES3 vs. ES2, $p = 1.82 \times 10^{-14}$<br>HS3 vs. HS2, $p = 1.79 \times 10^{-14}$ |
| Σ SFA | | | 19.94 | 21.33 | 24.12 | 24.97 | |
| Σ MUFA | | | 28.05 | 30.01 | 33.94 | 35.13 | |
| Σ PUFA | | | 13.53 | 14.48 | 16.37 | 16.95 | |
| n–3 | | | 10.05 | 10.76 | 12.16 | 12.59 | |
| n–6 | | | 3.19 | 3.42 | 3.86 | 4.00 | |
| n–3/n–6 | | | 3.15 | 3.15 | 3.15 | 3.15 | |
| n–6/n–3 | | | 0.32 | 0.32 | 0.32 | 0.32 | |

**Table 7.** *Cont.*

| Fatty Acid | Descriptive Statistics | Summer 2 | | Summer 3 | | *p* Values |
|---|---|---|---|---|---|---|
| | | Epaxial Muscles | Hypaxial Muscles | Epaxial Muscles | Hypaxial Muscles | |
| PUFA/SFA | | 0.68 | 0.68 | 0.68 | 0.68 | |
| USFA/SFA | | 2.09 | 2.09 | 2.09 | 2.09 | |
| PI | | 7.01 | 7.50 | 8.48 | 8.77 | |
| AI | | 0.57 | 0.57 | 0.57 | 0.57 | |
| TI | | 0.38 | 0.39 | 0.39 | 0.39 | |
| HFA | | 17.33 | 18.55 | 20.97 | 21.71 | |
| hFA | | 33.01 | 35.32 | 39.94 | 41.34 | |
| h/H | | 1.90 | 1.90 | 1.90 | 1.90 | |

SD—standard deviation. CV—coefficient of variation. ES2—epaxial muscles, summer 2. ES3—epaxial muscles, summer 3. HS2—hypaxial muscles, summer 2. HS3—hypaxial muscles, summer 3. PI: polyunsaturated index. TI: thrombogenic index. AI: Atherogenic Index. HFA: Hypercholesterolemic Fatty Acids (C12:0 + C14:0 + C16:0). hFA: hypocholesterolemic Fatty Acids (C18:1 + polyunsaturated FA). h/H: hypocholesterolemic/hypercholesterolemic FA. *n* = 7.

### 3.3. Amino Acids Profiling and the Nutritional Assessment of Polyodon Spathula Sturgeons Meat

The amino acids content of *Polyodon spathula* sturgeons fillets, aged 2 and 3 summers is displayed in Table 8.

**Table 8.** Amino acids content (g/100 g) in the fillets sampled from *Polyodon spathula* sturgeons aged two and three summers.

| Amino Acids | Summer 2 | | | Summer 3 | | | *p* Values |
|---|---|---|---|---|---|---|---|
| | Mean | ±SD | CV | Mean | ±SD | CV | |
| Tryptophan | 0.202 | 0.005 | 2.65 | 0.204 | 0.006 | 2.95 | >0.9999 |
| Threonine | 0.790 | 0.017 | 2.18 | 0.799 | 0.028 | 3.50 | 0.9999 |
| Isoleucine | 0.830 | 0.035 | 4.21 | 0.839 | 0.035 | 4.13 | 0.9999 |
| Leucine | 1.463 | 0.049 | 3.37 | 1.480 | 0.052 | 3.53 | 0.9924 |
| Lysine | 1.654 | 0.051 | 3.09 | 1.673 | 0.053 | 3.14 | 0.9628 |
| Methionine | 0.533 | 0.023 | 4.25 | 0.539 | 0.019 | 3.60 | >0.9999 |
| Cysteine | 0.193 | 0.008 | 4.19 | 0.195 | 0.007 | 3.48 | >0.9999 |
| Phenylalanine | 0.703 | 0.026 | 3.67 | 0.711 | 0.024 | 3.41 | 0.9999 |
| Tyrosine | 0.608 | 0.020 | 3.28 | 0.615 | 0.026 | 4.16 | 0.9999 |
| Valine | 0.928 | 0.027 | 2.94 | 0.939 | 0.035 | 3.68 | 0.9998 |
| Arginine | 1.077 | 0.036 | 3.38 | 1.090 | 0.047 | 4.32 | >0.9999 |
| Histidine | 0.530 | 0.011 | 2.13 | 0.536 | 0.022 | 4.05 | 0.9997 |
| Alanine | 1.088 | 0.046 | 4.21 | 1.101 | 0.036 | 3.24 | 0.8220 |
| Aspartic acid | 1.843 | 0.053 | 2.86 | 1.865 | 0.051 | 2.73 | 0.1275 |
| Glutamic acid | 2.688 | 0.095 | 3.55 | 2.719 | 0.091 | 3.34 | 0.9999 |
| Glycine | 0.864 | 0.035 | 4.01 | 0.874 | 0.038 | 4.34 | 0.9999 |
| Proline | 0.637 | 0.024 | 3.69 | 0.644 | 0.022 | 3.47 | 0.9999 |
| Serina | 0.735 | 0.023 | 3.18 | 0.744 | 0.023 | 3.05 | >0.9999 |
| ∑ amino acids | 17.365 | | | 17.568 | | | |
| ∑ essential amino acids | 7.102 | | | 7.185 | | | |
| ∑ semi-essential amino acids | 0.801 | | | 0.810 | | | |
| ∑ non-essential amino acids | 9.463 | | | 9.573 | | | |

In order to nutritionally evaluate the investigated fish proteins, the content of amino acids was calculated as g amino acids/100 g protein, so that the obtained data can be compared with the FAO/WHO standards for different categories of consumers (Table 9).

**Table 9.** Amino acids content (g/100 g protein) of the etalon protein and of the fillets sampled from *Polyodon spathula* sturgeons, aged two and three summers.

| Amino Acids | FAO/WHO Etalon Protein | | | S2 | S3 |
|---|---|---|---|---|---|
| | Standard 1 Children | Standard 2 Youth | Standard 3 Adults | | |
| Tryptophan | 1.7 | 1.0 | 0.6 | 1.1 | 1.1 |
| Threonine | 4.3 | 4.0 | 2.6 | 4.4 | 4.4 |
| Isoleucine | 4.6 | 4.0 | 3.0 | 4.6 | 4.6 |
| Leucine | 9.3 | 7.0 | 4.4 | 8.1 | 8.1 |
| Lysine | 6.6 | 5.5 | 3.1 | 9.2 | 9.2 |
| Methionine + Cystine | 4.2 | 3.5 | 2.7 | 4.0 | 4.0 |
| Phenylalanine + Tyrosine | 7.2 | 6.0 | 3.3 | 7.3 | 7.3 |
| Valine | 5.5 | 5.0 | 2.3 | 5.2 | 5.2 |
| EAA (g/16 g N) | 43.4 | 36 | 22 | 43.9 | 43.9 |

S2—2nd summer. S3—3rd summer. EAA (g/16 g N)—exogenous amino acids.

Nutritional evaluation of the proteins in the meat of *Polyodon spathula* sturgeons, aged two and three summers, is presented in Table 10.

**Table 10.** Nutritional assessment of the proteins in the *Polyodon spathula* sturgeons meat, aged two and three summers.

| Amino Acids | S2—Chemical Indices | | | S3—Chemical Indices | | |
|---|---|---|---|---|---|---|
| | Standard 1 | Standard 2 | Standard 3 | Standard 1 | Standard 2 | Standard 3 |
| Tryptophan | 66.00 | 112.20 | 187.00 | 65.88 | 112.00 | 186.67 |
| Threonine | 102.07 | 109.73 | 168.81 | 102.05 | 109.70 | 168.77 |
| Isoleucine | 100.24 | 115.28 | 153.70 | 100.15 | 115.18 | 153.57 |
| Leucine | 87.40 | 116.11 | 184.73 | 87.39 | 116.10 | 184.70 |
| Lysine | 139.23 | 167.07 | 296.42 | 139.20 | 167.04 | 296.35 |
| Methionine + Cystine | 96.02 | 115.23 | 149.37 | 95.98 | 115.17 | 149.30 |
| Phenylalanine + Tyrosine | 101.15 | 121.38 | 220.70 | 101.14 | 121.37 | 220.67 |
| Valine | 93.75 | 103.12 | 224.17 | 93.76 | 103.14 | 224.22 |
| EAAI (%) | 96.43 | 118.81 | 193.47 | 96.40 | 118.76 | 193.39 |
| BV | 93.41 | 117.80 | 199.18 | 93.38 | 117.75 | 199.10 |
| NI (%) | 17.36 | 21.39 | 34.82 | 17.55 | 21.63 | 35.22 |

S2—2nd summer. S3—3rd summer. EAAI (%)—essential amino acids index. BV—biological value. NI (%)—nutritional index.

## 4. Discussion

### 4.1. Chemical Proximate Composition

The most valuable part of a fish, the musculature, reaches up to 40–50% of its weight and includes: mostly the trunk musculature (the lateral muscles-the fillet, the red muscles and the muscles of the odd fins); head musculature (mandible and gill muscles); musculature of the girdles and of the even fins [1].

Water content of the fillet muscles in *Polyodon spathula*, varied between 75.65% in S3 and 77.83% in S2 (Table 3); water having a variable content depending on the age of the fish (higher at younger ages); the assessed values felt within the limits specified by literature [38,39]. From a statistical point of view, high significant differences were revealed between the two age categories ($p < 0.001$) (Table 3).

The proportion of proteins in the fillet of the studied fish ranged between 17.94% and 18.01%, data which are similar to those in other studies [38,65]. The assessed protein content places this species in the group of protein fish (15–20% protein) [66,67]. Even in the case of protein content, no significant statistical differences were found between the two ages ($p > 0.05$) (Table 6).

In general, in fish, the lipid content varies within very wide limits (0.1–28%) [43,66], fish being classified in: fatty fish, with more than 8% fat; fish with medium fattening status, between 4 and 8%; lean fish, with less than 4% fat [42,67]. In the conditions of our study, when no supplemental feeding was directly provided to paddlefish, the fillets had a lipid content between 2.82 and 3.98%, values that place the analyzed sturgeons in the class of fish with low lipid content. In comparison with other original findings on paddlefish aged one summer [42,67], lipid level in summer two and three was higher by 15–62%, following the natural trend of lipid accumulation as body reserves in parallel with fish ageing. In this situation, the obtained values obtained felt within the limits specified in the literature [38,65,68,69]. There are authors stating that meat of cultured sturgeons should have a lipid content between 5 and 10% [70]. High significant differences ($p < 0.001$) were found for the fillet total fat between the two analyzed ages (Table 6).

In comparable studies [71], data related to proximate composition of other sturgeon species revealed meat moisture content of 75.5% in *Acipenser baerii* (Siberian sturgeon) and of 77.7% in *Acipenser transmontanus* (White sturgeon), while in our findings, the fillet had 76.6–77.8 water content. Related to ash level, paddlefish samples from our findings ranged between 1.2–1.4%, while in the Siberian sturgeon it reached 1.3% and in White sturgeon 1.1%. Paddlefish meat analyzed in our study was lighter in terms of lipids accumulation (2.8–3.9%), in comparison with the Siberian sturgeon (5.6%) and richer than the White sturgeon (2.6%). Total protein content was pretty similar (17.9–18.1%), in comparison with 17.6% (*A. baerii*) and 18.6% (*A. transmontanus*).

Among the most present and most mechanically and enzymatically degrading resistant proteins in the connective tissues, collagen participates in maintaining the structural integrity of the tissues. From a chemical point of view, collagen is an incomplete protein, with low biological value [42,43]. The percentage of collagen in the fillets sampled from the studied fish was 3.93% in the third summer and 4.04% in the fourth summer. The collagen content is 3–10% for most fish species, and in the case of homoeothermic animals, it can reach up to 17% of the total protein [42,43,66,67]. So, in the analyzed sturgeons, the proportion in collagen fell within the quoted values. Statistically, no significant differences were found between the two ages ($p > 0.05$) (Table 6).

Water-protein ratio (W/P) is a criterion for evaluating the quality of fish meat. Accordingly, fish are ranked into five categories: 1st—fish with high nutritional value (W/P = 2.5–3.5); 2nd—fish with good nutritional value (W/P = 3.5–4.2); 3rd—fish with mediocre nutritional value (W/P = 4.2–4.7); 4th—fish with low nutritional value (W/P = 4.7–5.2) and 5th—fish in a state of advanced starvation (W/P above 5.2) [42,43,67]. Studied *Polyodon spathula* sturgeons felt into the category of fish with mediocre food value (3rd). In previous studies [43], W/P ratio has been found to be 4.33 (mediocre) in paddlefish aged one summer and 3.79 (good nutritional value) in paddlefish aged four summers, suggesting that meat becomes more qualitative starting with summer two.

Gross energy value of the fillet from paddlefish aged two summers was 102.05 kcal/100 g and of those aged three summers was 113.22 kcal/100 g (+10.94% energy, compared to previous year) ($p < 0.001$), suggesting an increase of nutritional value consequently to ageing but also with body development, a fact that is mainly due to the increase of lipids amount in the meat. The meat issued from three summers aged paddlefish was richer in calories than the meat produced by other (105 kcal/100 g) [69–71].

Assessment of the chemical composition of epaxial muscles group (dorsal ED and costal EC) and of the hypaxial muscle group (costal HC and abdominal HA) was carried on to identify the occurrence of any differences between the groups. In the paddlefish aged two summers, water content was quite low HA muscles (44.05%) compared to the ED, EC and HC muscles, which had 56.21–57.1% lower values. In all comparisons run between the four muscles, only the costal ones (EC and HC) did not differ significantly ($p > 0.05$) for the water and total solids content, while the other differences were highly significant ($p < 0.001$) (Table 4). The HA muscles had a much higher energy value (more than 400 kcal by 100 g, 3.87–4.29 folds compared to other muscle groups) ($p < 0.001$) due to the much

higher presence of lipids in the abdominal muscles compared to the other muscle groups ($p < 0.001$). These parameters also influence the W/P ratio which was lower for HA muscles (2.63), by 59.37–61.88% compared to ED, EC and HC muscles ($p < 0.001$) (Table 4).

In the samples issued from *Polyodon spathula* sturgeons aged three summers, there were high significant differences ($p < 0.001$) found for water and total solids content between muscles ED, EC, HC, and HA (the first three muscle groups comprise 57.15–57.43% more water than the HA muscles). On the contrary, the HA muscles were 8.51–11.34% richer than the other analyzed muscles, that led to a gross energy content also higher in HA ($p < 0.001$) and to a much lower W/P ratio (by 60.8–66.2%), compared to the other three other studied muscle groups ($p < 0.001$) (Table 5).

When the influence of fish age on the proximate composition of the meat was analyzed (Table 6), only the total fat content presented significant differences ($p < 0.001$), particularly for the EC and HC muscles, while the gross energy content varied significantly ($p < 0.001$) in relation with age (S3 vs. S2) in ED, EC, and HC muscles. Therefore, fish ageing induces excessive lipids accumulation not only on filled but also on other muscle groups, hence the higher gross energy meat content in summer three, compared to summer two (especially in those muscles having less caloric values in summer two, due to lower lipid content, in comparison with HA muscles, which accumulated fat earlier before summer three).

*4.2. Fatty Acids Profiling and the Sanogenic Indices of the Polyodon Spathula Sturgeons Meat*

According to the data in Table 7, among the 20 identified fatty acids, Palmitic acid had the highest occurrence (15.31–19.18 g/100 g total fatty acids), closely followed by the Oleic acid (15.18–19.01 g/100 g total fatty acids), a fact also highlighted by other authors for paddlefish meat [39,41] or in the case of other sturgeons [71–74].

Out of the total of fatty acids identified in the analyzed samples, the highest proportion was taken by the MUFA (approx. 45.6%), followed by the SFA (approx. 32.4%), then by PUFA (approx. 22%), which indicates the presence of good quality fat in the meat of *Polyodon spathula* sturgeons aged two and three summers.

The high content of MUFA and PUFA, known to have a beneficial effect on human health (especially due to their protective role against cardiovascular diseases, as well as the values for the cholesterol content [39,41]) make the meat of the paddlefish an important source of "good fats". The degree of assimilation of fish fats, in human consumers compared to other fats, is very high, a fact explained, first of all, by the particular presence of linoleic, linolenic, arachidonic, eicosapentaenoic, docosapentaenoic, and docosahexaenoic acids. Nutritionally, the increased n–3/n–6 PUFA ratio occurring in *Polyodon spathula* sturgeon meat may also have a protective effect against breast cancer [75].

In other sturgeon species, total SFA reached 19.2 g/total FA (*Acipenser baerii*)—close to our findings for two summers paddlefish and 24.5 g/total FA (*Acipenser transmontanus*), comparable to the samples in our study issued from summer three individuals [71].

As expected, the analysis of meat proximate composition revealed increasing accumulation of lipids from the 2nd to the 3rd summer. Consequently, MUFA increased from 28.05 to 33.94% of FAME (epaxial muscles) and from 30.01 to 35.13% of FAME (hypaxial muscles). Total MUFA varied between 45.5 g/total FA in *A. baerii* and 31.3 g/total FA in *A. transmontanus*, both species richer than the samples from *P. spathula*. Siberian sturgeon meat and Asian sturgeon meat were two to three folds richer in PUFA (35.3 g/total FA–44.2 g/total FA), compared to PUFA levels found in paddlefish investigated meat [71].

The high values of the polyunsaturation index (PI) of the II and III summer paddlefish meat (7.01–8.77) indicate the high level of PUFA, an aspect considered important for human health due to the implications in regulating the level of blood cholesterol [76,77].

From the perspective of human health, the thrombogenic index (TI) and atherogenic index (AI) highlight the predisposition to the incidence of cardiovascular diseases and express the relationship between saturated (pro-thrombo/atherogenic) and unsaturated (anti-thrombo/atherogenic) lipids [52,78]. The AI value calculated for sturgeon *Polyodon spathula* aged 2 or 3 summers (AI = 0.57) is similar to those of carp (0.57) [79] and with

approx. 14% lower of that found in trout (0.65) [80]. The highest AI values were reported by Kucukgulmez et al. [80] in two saltwater fish species (AI = 1.22). The TI values calculated for the studied sturgeons were very low compared to other freshwater fish species (carp TI = 0.63, trout TI = 0.49) [79], which reveals a very low tendency for blood clotting in those consuming such meat.

*Ployodon spathula* sturgeon meat is characterized by the fairly high presence of fatty acids with hypocholesterolemic effects (hFA) (33.01–41.34), an aspect that also results from the high h/H ratio (hypocholesterolemic/hypercholesterolemic FA) (1, 9). The value of the h/H index suggests the presence of sufficiently valuable lipids, with the potential to lower consumers' blood plasma cholesterol [55].

### 4.3. Amino Acids Profiling and the Nutritional Assessment of Polyodon Spathula Sturgeons Meat

Data in Table 8 reveal slightly higher values of amino acids content in three summers aged sturgeons, in close correlation with the dynamics of meat water and protein content. The most abundant amino acids were Glutamic acid (2.688% in S2 and 2.719% in S3), Aspartic acid (1.843% in S2 and 1.865% in S3), Lysine (1.654% in S2 and 1.673% in S3), followed by Leucine, Alanine, Arginine, Valine, Glycine, Isoleucine, Threonine, and by Tryptophan (0.202% in S2 and 0.204% in S3). The sum of amino acids was quite similar in summers 2 and 3 compared to previous original research results where levels of 17.44 g/100 g total amino acids were reported [43]. The amino acids on the first three positions had the same sequence in the case of the data reported by other authors [41,81,82]. The presence of Glutamic acid gives the meat of sturgeon *Polyodon spathula* a special taste. Also, the special role of this amino acid in brain metabolism has been highlighted since it participates in the synthesis of several physiological substances [83–87].

The analysis by category of amino acids (essential, semi-essential and non-essential) shows revealed that essential amino acids represent approx. 40.9%, the semi-essentials represent 4.61%, while the non-essentials ones represent 54.49%.

In order to estimate the biological value of the proteins from the meat of *Polyodon spathula* sturgeons, aged 2 and 3 summers by chemical methods, the content of essential and semi-essential amino acids (g/100 g protein) was assessed, then reported to the values specified within FAO and WHO standards for three categories of consumers (Table 10). These comparisons revealed high values for all amino acids and for both ages of fish in youth and adult consumers; for children the requirements for Tryptophan, Valine and Methionine + Cystine were not covered.

By calculating the Oser index or EAA index [43,48,63], it was possible to show that these fish contain high quality protein since the proportion of essential and semi-essential amino acids passed above 100% of etalon protein (118.76–193.47%) in the case of young people and adults. On the contrary, for children, the values were slightly lower, compared to etalon protein (96.40% in S3 meat and 96.43% in S2 meat), suggesting an insufficient degree of coverage of the essential amino acid requirements for this category of consumers. This fact was also highlighted by the calculation of BV and NI (%).

The obtained results are consistent with other reported data on the amino acid content and nutritional value of the meat of *Polyodon spathula* sturgeons [41,88–90], as well as of other sturgeons [91].

## 5. Conclusions

Our study brings scientific novelty through a detailed approach of paddlefish meat nutritional quality, of detailed muscle groups, supported with particular analysis of protein biological values in comparison with international standards for different age groups of consumers, as well as of sanogenic indices derived from the lipidic profile of the meat.

Proximate chemical composition analysis of the meat from *Polyodon spathula* sturgeons, aged two and three summers highlighted that between the two ages there are no significant differences for the chemical composition of the fillets, whereas significant differences occurred between muscle groups. In the case of both ages, there were significant differences

between HA muscles and the other three analyzed groups (HC, ED, and EC), mainly due to the accumulation of fat throughout one year or growth. This fact has also significantly impacted the gross energy content of the analyzed meat.

Nutritional evaluation of the paddlefish meat indicated that the fats are of good quality with a significant presence of PUFA, and good values for sanogenic indices were found as well. Also, the protein quality is good for youth and adults and good enough for children.

According to our findings, the optimal age to capitalize paddlefish meat is the 3rd summer, when the best meat yields, proximate composition and nutritional value were met. The study mainly focused on these nutritional aspects and less on the technological factors related to quantitative productions. It would be interesting to overcome such shortcomings in future studies, taking into account all factors and building-up, eventually, a mathematical model to quantify the degrees of influence on quantitative and qualitative paddlefish meat production.

As a follow-up research project, it would be interesting to run textural instrumental analysis on paddlefish meat, accompanied by an analysis of the influence of cooking methods on technological and nutritional quality, in parallel with sensory evaluation and public tasting campaigns to support the degree of awareness and acceptability in consumers for paddlefish (knowing that consumers do not prefer this kind of meat despite its high nutritional features).

**Author Contributions:** Conceptualization, C.S. and D.S.; methodology, D.S., O.S.M. and R.-M.R.-R.; software, D.S. and R.-M.R.-R.; validation, C.S. and R.-M.R.-R.; formal analysis, O.S.M. and D.S.; investigation, C.S., O.S.M. and D.S.; data curation, C.S. and R.-M.R.-R.; writing—original draft preparation, D.S. and C.S.; writing—review and editing, D.S. and R.-M.R.-R.; visualization, D.S. and R.-M.R.-R.; supervision, D.S. All authors have read and agreed to the published version of the manuscript.

**Funding:** This research received no external funding.

**Institutional Review Board Statement:** Not applicable. No animals were used for applying experimental factors on them and to measure their effect throughout certain specific reasoning criteria. The fish individuals used for meat sampling were raised in common with the other marketable fishes in the farm and the samples were taken after the exitus was installed in refrigeration marketing condition.

**Informed Consent Statement:** Not applicable.

**Data Availability Statement:** Data supporting reported results available, upon request, at the authors.

**Conflicts of Interest:** The authors declare no conflict of interest.

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
