# Peer review of "Qualitative and Nutritional Evaluation of Paddlefish (Polyodon spathula) Meat Production"

_agriculture, doi:10.3390/agriculture12111965_

Round 1

Reviewer 1 Report

The authors compare the qualitative composition of paddlefish aged two and three summers, analyzing different chemical parameters and also nutritional value and quality.

The work is well written with a lot of data comparing the two years and highligthed no significatively difference in fillets between the two ages.

Material and methods are well described , and statistical analysis done.

Results and discussion seem to be complete, and the paper present  good number of references.

Some inaccuracies:

Line 259, space (amino acid/g protein)

Check the bibliographic references, year, volume are written now in bold, now normal, now italic, double check.

More comments please see the attached files.

Author Response

Honorable reviewer,

Thank you indeed for taking time in evaluating our manuscript and providing us appreciations and also useful and insightful suggestions that will improve it.

We will answer to your recommendations /suggestions punctually here and by commenting directly in the revised manuscript.

Some inaccuracies:

Rec. 1> Line 259, space (amino acid/g protein) -

Solved, thank you!

Rec. 2 Analyzing the tables I can suggest to apply the statistical analysis also to the data of fatty acid and amino acid which are described in a single value and without SD and Anova. 

Answer: tables with FA and AA were processed accordingly to add SD, CV and p values, thank you!

Rec. 3> we have checked the writing style of references and have synchronized it with the indications in the manuscript editing guidelines. Thank you!

Reviewer 2 Report

This manuscript is a continuation and in many respects a repetition of the research of the team of authors on the issue of Polyodon spathula sturgeon meat characteristics. In particular articles: Simeanu D., Creanga S., Simeanu C. Research on the meat quality produced by Polyodon Spathula sturgeons species related to 550 human nutritional requirements.  Research Journal of Biotechnology, 2015, 10, no.  6, pp.  36-43. Simeanu C.; Simeanu D.; Popa A.; Usturoi A.; Bodescu D.; Dolis, M.G. Research Regarding Content in Amino-acids and Biolog-552 ical Value of Proteins from Polyodon spathula Sturgeon Meat. Revista de chimie, 2017, Volume 68, Issue 5, pp. 1063-1069, 553 WOS:000405816300037, ISSN 0034-7752 etc.

In this regard, as well as directly the content of this work, there are several significant concerns.

1) The introduction duplicates large parts of previously published works.  At the same time, no information is provided about the characteristics of aquaculture of this species, previous studies in this area, the relevance of this study and its distinctive features.

2) The conditions and years of growing fish are not given. What were the climatic and other conditions in the region during these years, essential for growing fish? The feeding regimes used and the feeds used are not indicated.  This significantly limits the application, interpretation and reproducibility of the results.

3) Most of the obtained parameters correspond to the previously obtained data.  What is the supposed scientific novelty of this study?

4) This study examined fish of 2nd and 3d summer.  Previously, the authors studied fish from 1st to 4d summers.  A comparison of the data obtained for different years is not given.

5) Despite the fact that the article is devoted specifically to the nutritional characteristics of the production of fish meat, the production aspects are not considered.  What can affect the obtained characteristics in terms of production?  What season and year is more appropriate to use fish meat?  Etc.

6) Line 212 has a duplication of a word.

7) Vendors of materials and equipment are not given in the format company, city, country.

8) The organoleptic and structural-mechanical characteristics of fish meat have not been studied.

9) Possible anti-alimentary and biological value-reducing aspects of fish meat have not been investigated.

Thus, the main concerns are the scientific novelty of this study and its relevance to the specified title.  To eliminate them, a very deep refinement and processing of the manuscript is required.

Author Response

Honorable reviewer,

Thank you indeed for taking time in evaluating our manuscript and providing us appreciations and also useful and insightful suggestions that will improve it.

We will answer to your recommendations /suggestions punctually here and by commenting directly in the revised manuscript.

Rec. 1) The introduction duplicates large parts of previously published works.  At the same time, no information is provided about the characteristics of aquaculture of this species, previous studies in this area, the relevance of this study and its distinctive features.

Answer: some experimental data achieved from the samples analysed throughout the experimental period were published in terms of proximate composition, despite the fact the data in this article is newly acquired using preserved samples. Starting from the new original analytical findings we elaborated on the nutritional values of proteins - amino acids, in comparison with FAO/WHO standards and we analysed the fatty acid profile and cholesterol, developing the sanogenic indexes related mostly to the cardiovascular risks in consumers. Moreover, the results are presented on more detailed muscular groups and this would be also a degree of novelty.

Also, we considered your recommendations in adding in the paper supplemental data related to fishery technological aspects (polyculture, diets etc.), in order to complete the big picture of the experimental conditions. Thank you!

2) The conditions and years of growing fish are not given. What were the climatic and other conditions in the region during these years, essential for growing fish? The feeding regimes used and the feeds used are not indicated.  This significantly limits the application, interpretation and reproducibility of the results.

Answer: we added in the paper supplemental data related to fishery technological aspects (region, climate conditions, polyculture structure, diets etc.), in order to complete the big picture of the experimental conditions. Thank you!

3) Most of the obtained parameters correspond to the previously obtained data.  What is the supposed scientific novelty of this study?

Answer: The novelty is the development of the proximate composition findings into nutritional data related to Amino Acids intake standards, Fatty acids profile and Sanogenic indices. Also, the results are issued from detailed muscle groups, in comparison with other previous studies. Thank you!

4) This study examined fish of 2nd and 3d summer.  Previously, the authors studied fish from 1st to 4d summers.  A comparison of the data obtained for different years is not given.

Answer: Thank you. We inserted in discussion section some comparative results from, issued in our previous research.

5) Despite the fact that the article is devoted specifically to the nutritional characteristics of the production of fish meat, the production aspects are not considered.  What can affect the obtained characteristics in terms of production?  What season and year is more appropriate to use fish meat?  Etc.

Answer: Thank you! Basing on the technological apects in the farm, it is recommended to capitalise the production by the end of the 3rd summer, when there is the best ratio between technological investments and product quality.

6) Line 212 has a duplication of a word.

Answer:  solved, thank you!

7) Vendors of materials and equipment are not given in the format company, city, country.

Answer: solved, thank you

8) The organoleptic and structural-mechanical characteristics of fish meat have not been studied.

Answer: the purpose of the study was to deepen the proximate composition in relation with nutritional and sanogenic indices, in comparison with FAO/WHO standards for different categories of consumer. Textural analysis was not performed on the samples and we cannot introduce such data. Thank you! Your recommendations will be put in the paper conclusive part as follow-up of further studies on the same species. Thank you! 

9) Possible anti-alimentary and biological value-reducing aspects of fish meat have not been investigated.

Answer: we did not carry on analytical tests related to the identification and quantification of anti-alimentary and value-reducing molecules existing in samples, therefore we did not put such data as results. Your recommendations will be put in the paper as follow-up of further studies on the same species. 

Thank you!

Reviewer 3 Report

The reviewed manuscript “Qualitative and Nutritional Evaluation of Paddlefish (Polyodon spathula) Meat Production” is well written, and covers an interesting topic on Polyodon spathula meat evaluation. As a valuable species of sturgeon, Polyodon spathula meat has a well taste and texture, but it remains unacceptable to consumers and is limited in the market. Therefore, research on quantitative and qualitative productive performance for Paddlefish is of significant importance. The paper analyzed the chemical composition, cholesterol and collagen content of Paddlefish meat, as well as highlighted the nutritional value, the profile of fatty acids and amino acids. Those detailed data indicated that the fats are of high quality, and the protein quality is beneficial to youth and adults. However, the paper exists few defects which must be solved before it is considerable for publication.

1.In Introduction section, the author should refer to the current researches on qualitative and nutritional evaluation of paddlefish and other species of sturgeon as much as possible. Furthermore, it is recommended to supply detailed issues about deficiency of paddlefish quantitative and qualitative evaluation.

2. In Figure 1, the directions of 4 and 5 arrows are unclear. Please consider replacing them with clearer ones.

3. Obviously, there is an error in the serial number in the Discussion section.

4. Another obvious problem with this paper is the lack of sufficient analysis of the experiment results in Discussion part. The author needs to explain experiment results in detail by comparing with qualitative and nutritional evaluation of other sturgeons.

5. In Conclusions part, the author should not only explain the results, but also highlight the important findings, innovative contributions and afterthought of this work. The contribution of this article could be demonstrated by illustrating the shortcomings of previous researches. More importantly, the advantages and disadvantages of the paper should be analyzed and subsequent research directions should be proposed for reference.

6. In Table 7, the decimal places of standards and experiment results are inconsistent.

Author Response

Honorable reviewer,

Thank you indeed for taking time in evaluating our manuscript and providing us appreciations and also useful and insightful suggestions that will improve it.

We will answer to your recommendations /suggestions punctually here and by commenting directly in the revised manuscript.

Some inaccuracies:

Rec. 1> In Introduction section, the author should refer to the current researches on qualitative and nutritional evaluation of paddlefish and other species of sturgeon as much as possible. Furthermore, it is recommended to supply detailed issues about deficiency of paddlefish quantitative and qualitative evaluation.

Answer: we have inserted data related to other species of sturgeons both in introduction and discussion data, as comparation. Also we put a paragraph related to issues and defficiences on paddlefish quantitative and qualitative evaluation.

Rec. 2 In Figure 1, the directions of 4 and 5 arrows are unclear. Please consider replacing them with clearer ones.

Answer: adjusted the figure arrows and numbering to correspond to the right indications, thank you!

Rec. 3> Obviously, there is an error in the serial number in the Discussion section.

Answer: Renumbered it. Thank you!

Rec. 4. Another obvious problem with this paper is the lack of sufficient analysis of the experiment results in Discussion part. The author needs to explain experiment results in detail by comparing with qualitative and nutritional evaluation of other sturgeons.

Answer: we inserted comparative data issued from other sturgeons meat and synchronised the reference list, as well. Thank you!

Rec. 5. In Conclusions part, the author should not only explain the results, but also highlight the important findings, innovative contributions and afterthought of this work. The contribution of this article could be demonstrated by illustrating the shortcomings of previous researches. More importantly, the advantages and disadvantages of the paper should be analyzed and subsequent research directions should be proposed for reference.

Answer: we inserted supplemental paragraph in conclusion, addressing your recommendations, thank you!

6. In Table 7, the decimal places of standards and experiment results are inconsistent.

Answer: reduced decimals in our computation to a single one, to correspond to the standards. Thank you!

Round 2

Reviewer 2 Report

The MS looks better after revision. Thank you.

Reviewer 3 Report

After revision, the manuscript could be accepted by the journal now.